# Influence of the Bonding Boundary Conditions of Timber-Glass I-Beams on Load-Bearing Capacity and Stiffness

Mateja Držečnik *, Andrej Štrukelj and Miroslav Premrov *

Faculty of Civil Engineering, Transportation Engineering and Architecture, University of Maribor, Smetanova 17, 2000 Maribor, Slovenia; andrej.strukelj@um.si

* Correspondence: mateja.drzecnik@um.si (M.D.); miroslav.premrov@um.si (M.P.); Tel.: +386-2-22-94-343 (M.D.); +386-2-22-94-303 (M.P.)

**Abstract:** Structural glass plays an important role in modern architecture, interior design, and building design. It has earned this title primarily because of its properties, such as transparency and its importance in lighting a space. Glass is a challenging building material because of its unpredictability and fragile behaviour. Its fragility, and the way it disintegrates, are the main reasons for using glass in collaboration with timber. The aim of this study is to provide researchers with a more detailed analysis of the influence of the cross-section of I-beams made of timber and glass on the load-bearing capacity and stiffness of each element, based on the research carried out as a basis for such a study. Special attention is focused on analysing the influence of different bonding line types. Composite materials are usually made of a combination of several materials. The goal in making composites is to create a synergy between these materials and combine the good properties of each part of the component.

**Keywords:** timber-glass I-beams; timber; glass; adhesive



## 1. Introduction

Connecting the outdoor environment with the indoor living environment is one of the most important advantages of modern construction. It is well known that natural light has a positive impact on health and quality of life. The need for greater transparency, proper orientation of buildings, and the reduction of energy losses of the whole building play an important role in the design of buildings. Transparent building elements pose great challenges to architects and especially to engineers. Glass has been around for more than a thousand years, but only in the last 20 years has it begun to be used as an integral part of load-bearing structural elements, such as walls, columns, and beams. Despite its positive properties, such as transparency, strength and durability, its fragility must be highlighted and considered.

The material behaviour of glass is linearly elastic and brittle. The fracture toughness and thus the parameters relevant for the dimensioning of load-bearing glass components are strongly dependent on the degree of pre-damage and edge treatment (micro-cracks, defects, or molecular inclusions), the environmental conditions (humidity, temperature), the dimensions of the glass (area of the loaded glass and thus the probability of the presence of defects), the duration of the load, and the type of application (probability of possible damage under load). The behaviour of glass during load transfer and failure is largely described by the rules of fracture mechanics. The tensile strength of glass is much less than its compressive strength, which is high, so it can be used for both vertical and horizontal loads. However, because of its brittleness, special attention must be paid to the distribution of loads. The behaviour of soda-lime-silica glass shows properties similar to concrete in many physical aspects.

Timber has many advantages as a building material because it is light, durable, and ecological. Its density makes it an extremely strong and flexible material that can withstand

extreme forces for short periods of time without failing. At the same time, its structure gives it lightness and efficiency as an insulating material. Timber has excellent insulating properties and helps to reduce thermal bridging problems. This puts it ahead of its main competitors in the construction sector—concrete and steel.

Timber and glass are special materials with specific properties. Joining them is therefore a demanding task that requires a lot of specific knowledge about the material properties of both materials and the joining element—the adhesive. A good knowledge of their advantages and disadvantages is therefore essential. The bond between timber and glass cannot be completely rigid because the two materials behave very differently due to their different coefficients of thermal expansion and humidity. Table 1 shows the material properties of float glass compared to timber C30. Glass has a high modulus of elasticity of about 70 GPa, which is about six times higher than the modulus of elasticity of softwood in the grain direction. This figure tells us that glass is a relatively rigid material. Properly used glass elements can therefore make an important contribution to the stiffness of the overall structure. The problem that remains is the behaviour of the glass, which is almost linearly elastic until failure. The strength of the glass is highly dependent on the type of load. The compressive strength of glass is extremely high, usually between 700 and 900 MPa, which is about 40 times the strength of timber. The tensile bending strength depends on the type of glass, but generally ranges from 45 MPa for float glass to 150 MPa for chemically strengthened glass. Both values are much higher than the tensile strength of timber. The coefficient of thermal expansion $\alpha_T$ is of critical importance when using timber-glass composites. $\alpha_T$ values for glass, softwood, and hardwood are $0.9 \times 10^{-5}$ K$^{-1}$, $0.5 \times 10^{-5}$ K$^{-1}$, and $0.8 \times 10^{-5}$ K$^{-1}$, respectively. For composites exposed to high temperature effects, increased shear stresses may occur in the adhesives between the timber and glass elements, and this should be considered in the design.

**Table 1.** Material properties of float glass and softwood C30 [1].

| | Density $\rho$ [kg/m$^3$] | Compress. Strength $f_c$ [N/mm$^2$] | Tensile Bending Strength $f_{mt}$ [N/mm$^2$] | Modulus of Elasticity E [N/mm$^2$] | Coeff. of Thermal Expansion $\alpha_T$ [$10^{-5}$ K$^{-1}$] |
|---|---|---|---|---|---|
| Float glass | 2500 | 800 | 45 | 70,000 | 0.90 |
| Timber C30 | 460 | 23 | 30 | 12,000 | 0.50 |
| Ratio glass/timber | 5.43 | 34.78 | 2.9 | 5.83 | 1.80 |

The concept of timber-glass I-beams (TGIB) emerged in the early 2000s. In the development of this type of beam, two concepts were presented in principle, namely TGIB with flanges attached directly to the glass web (TGIB-gw), and TGIB with groove in the flanges (TGIB-gf).

The first more technically sophisticated designs were studied by researcher Hamm [2] at the École Polytechnique Fédérale de Lausanne in Switzerland. He studied the impact of timber-glass I-beams with flanges attached directly to the glass web using polyurethane adhesive. He saw the advantage of such beams mainly in terms of their transparency or light transmission through the room, and their attractive appearance. Eight specimens were tested to the failure limit in a four-point bending test. The glass web was 10 mm thick, and the timber flanges varied from $30 \times 50$ mm to $50 \times 60$ mm. The specimens were 4000 mm long and 250 mm high. The reason for choosing this design was to transfer the loads from the timber to the glass via a suitable adhesive. All eight samples tested showed similar behaviour. The tests showed that timber and glass act as a homogeneous material in these composites. The timber even acts as a reinforcement for the glass, increasing the load-bearing capacity of the entire element when the first crack occurs. He observed up to a 200% increase in load from the first crack to the ultimate failure of the beam, which is called post-breakage reinforcement. Kreher [3,4] has written extensively on this topic. In 2004, he introduced the concept of timber-glass I-beams, following the example of J. Hamm. The cross-section consisted of timber flanges of different sizes, the lower $30 \times 30$ mm and

the upper 50 × 50 mm, which he glued together with polyurethane. For the glass web he used three different types of glass: annealed float glass, heat strengthened float glass, and fully temperate glass, with thicknesses of 4 mm and 6 mm. From a materials testing point of view, these thicknesses are sufficient, mainly because of the smaller range of samples tested, but for everyday use he recommended thicker glass. In this case, the specimens with annealed float glass also showed a 70% increase in load-bearing capacity after the initial crack and ductile fracture. This phenomenon, of course, was not observed with tempered and fully tempered glass. Kreher's beam concept was tested by Julius Natterer on a natural scale. Due to the good results of the tests, they were installed as roof beams in the conference hall of the Hotel Palafitte in Switzerland, which was built as part of the Swiss national exhibition Expo 0.2 in 2002. The built-in beams supporting the lightweight roof are designed to transfer snow and wind loads to the steel columns embedded in the exterior walls. The length of all the I-beams installed was 6000 mm and the height was 580 mm. Twelve mm thick single tempered glass was used. The upper timber flanges were made from two rectangular wooden blocks of 100 × 160 mm and the lower flanges from two smaller wooden blocks of 65 × 65 mm. The researcher justified the choice of these dimensions based on fire safety and load transfer. The dimensions of the upper flanges ensure that all loads are transferred even in the event of a complete failure of the glass, which in this case has no post-critical load bearing capacity after the initial crack. Samples of these beams were previously tested in a four-point bending test, in which they were subjected to a load three times higher than the maximum external load envisaged in the hotel. In 2008, Portuguese researchers Cruz and Pequeno [5,6] studied and tested 15 timber-glass I-beams. All beams were 550 mm high and varied in length from 650 mm to 3200 mm. The difference from previous studies was the use of laminated 6 mm annealed float glass panes with a PVB film in between. The timber flanges had a cross section of 70 × 100 mm. During testing, they also found that the load bearing capacity of a 3500 mm composite beam with polymer adhesive increased by up to 85% after the first crack. They also observed fewer cracks in the beam than previous researchers, mainly due to the use of a more rigid adhesive. The Portuguese researchers were also among the first to study the adhesives that hold timber-glass composites together. They divided them into three groups, namely: rigid adhesives—extremely high strength and stiffness (e.g., acrylate, epoxy), medium-stiff adhesives—balanced strength and elasticity (e.g., polyurethane) and elastic adhesives—extreme elasticity and low strength (e.g., silicone and some polyurethanes). Finally, they recommended silicone as the most suitable adhesive for bonding timber-glass composite beams because of its elasticity. They also recommended that the higher elasticity and associated lower stiffness of adhesives warrants thorough investigation, especially when adhesives are used in load-bearing structures. In such a composite, the timber provides the ductility, and the glass provides the resistance and stiffness. Research on I-beams made of timber and glass continued, and in 2008–2011 the research results of Swedish researchers Blyberg et al. were presented [7]. A slightly different model than the previous ones was performed. All beams were 3850 mm long and 240 mm high. The beams were made of laminated veneer lumber (LVL) with dimensions of 45 × 60 mm. The special feature of this beam cross-section was the groove into which they glued a glass web using different adhesives. Two different groove widths of 13 mm and 15 mm were tested. Two different edge finishes for the glass web were also tested. For comparison purposes, the beam was made with a silicone adhesive. A similar concept of the beam, as seen in the work of Swedish researchers, was published by Kozlowski and Hulmika in 2013 [8]. The Polish researchers used an 8 mm thick annealed float glass web and 55 mm × 75 mm timber flanges. The length and height of the test specimens were 1800 mm × 300 mm. Kozlowski et al. [9,10] published another study in 2013 that will serve as a comparison to our experiments. This study will be presented in more detail later in the article. Over the years, more research has been done on TGIB [11–30]. From all these studies, two main types of TGIBs can be derived, namely, TGIBs with flanges directly attached to the glass web, and TGIBs with grooves in the flanges.

In this paper, a detailed comparison between our TGIBs with flanges directly attached to the glass web (TGIB-fw) and TGIBs with grooves in the flanges (TGIB-gf) from the researcher Kozlowski et al. was presented. In TGIB-fw, the glass web was not fully integrated into the timber flange, which allowed for greater temperature expansion between the two materials. It was expected that the ductility of such elements is higher, but probably also has a lower bending stiffness and load-bearing capacity, which we aimed to demonstrate in this study. The main novelty of this study lies in the connection of the glass web to the timber flanges and the comparison with another connection method—TGIB-gf, in particular the study of Kozlowski et al. [10], where a slightly lower ductility was expected. The temperature effect was not considered in the study.

First, the theoretical background was given, then basic data of both cross sections were compared, and then the design of the specimens included in the experimental analysis was analysed. Both concepts were evaluated using the load-displacement diagrams obtained from the bending tests and finally the results and recommendations were presented.

## 2. Theoretical Background

*Gamma ($\gamma_i$) Method*

The effective bending stiffness of timber glass composite beams depends on the cross-section parameters and the connection stiffness—gamma factor ($\gamma_i$) between the individual layers. In the case of mechanical connectors, the value of the factor depends on the ratio between the distance between the connectors s and the slip modulus of the connectors [31]. For the rigid connection, as it is the bonded connection, $\gamma_i = 1.0$, for no mechanical connection between individual layers $\gamma_i = 0$.

The effective bending stiffness according to Figure 1 is calculated as:

$$(EI)_{eff} = \sum_{n=1}^{3} E_i I_i + \gamma_i \cdot E_i \cdot A_i \cdot a_i^2 \tag{1}$$

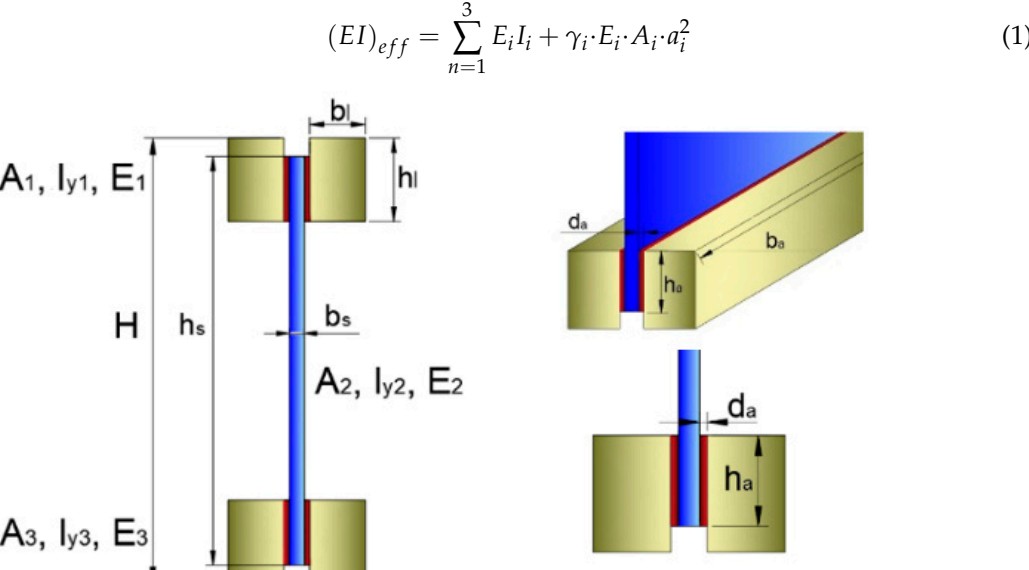

**Figure 1.** Schematic drawing of the symbols for the calculation of TGIB-fw.

Thus, the $\gamma$-factor for the adhesive in our case depends on the properties of the flanges ($E_1 = E_2$ and $A_1 = A_2$), the total length of the beam itself, and the slip modulus of the adhesive $K_k$ written as:

$$K_k = \frac{K_{1(3)}}{s_{1(3)}} \tag{2}$$

We start from a mechanical model [2]. For the shear stress we can write:

$$\tau_{xy} = G \cdot \left[ \frac{\partial u}{\partial y} + \frac{\partial v}{\partial x} \right] \tag{3}$$

Referring to Figure 2 and assuming small displacements and constant shear strains through the thickness of the adhesive joint, the relative displacements between timber and glass can be described as follows:

$$\delta = d_a \cdot \frac{\partial u}{\partial y} \quad \rightarrow \quad \frac{\partial u}{\partial y} = \frac{\delta_{(x)}}{d_a} \tag{4}$$

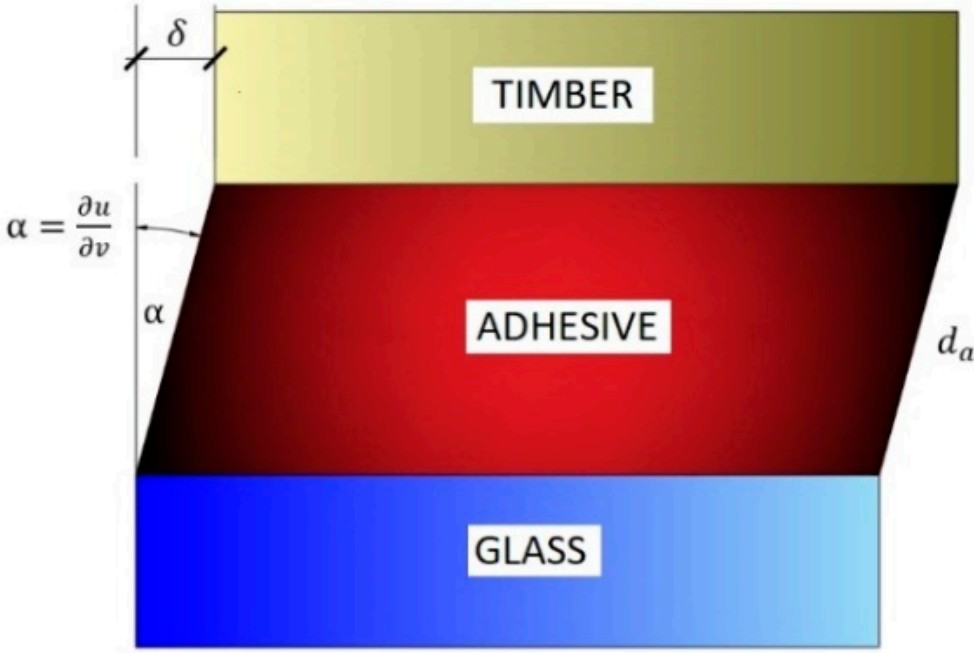

$$\alpha = \frac{\partial u}{\partial v}$$

**Figure 2.** Shear deformation of an adhesive joint.

Establish the equilibrium and write:

$$0 = dF_s - \tau_{xy} \cdot b_a \cdot 2 \cdot dx \tag{5}$$

We assume a parallel path between the timber and the glass, so we can neglect the y-direction. Apply Equations (3) and (4) and obtain for the glass:

$$dF_s = 2 \cdot \frac{b_a}{d_a} \cdot G \cdot \delta_{(x)} \cdot dx \tag{6}$$

For an infinitesimal element, $\delta$ can be assumed to be constant along the length of the element in the x-direction, since the stiffness of timber and glass with respect to the adhesive is extremely high.

$$\Delta F_s = 2 \cdot \frac{b_a \cdot l}{d_a} \, G \cdot \delta \tag{7}$$

We can also write it in a slightly different form:

$$k_{com} = 2 \cdot \frac{b_a \cdot l}{b_a} \cdot G \cdot \delta \quad \rightarrow \quad \Delta F_s = k_{com} \cdot \delta \tag{8}$$

With reference to Figures 2–4, we can now write:

$$dF_s = 2 \cdot \tau_{xy} \cdot b_a \cdot dx \tag{9}$$

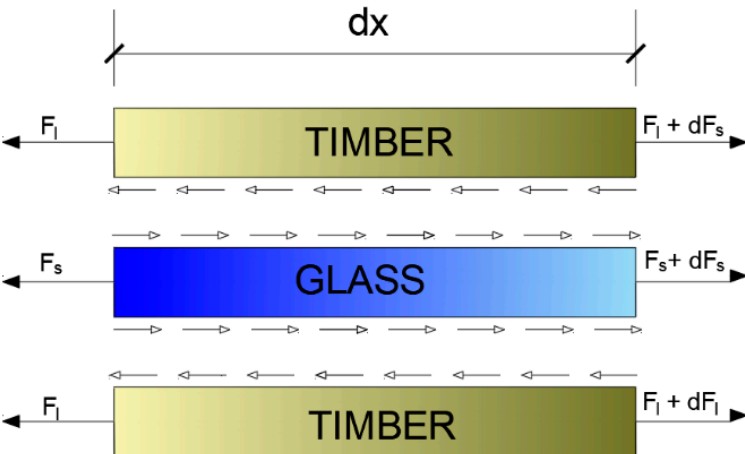

**Figure 3.** Differential element.

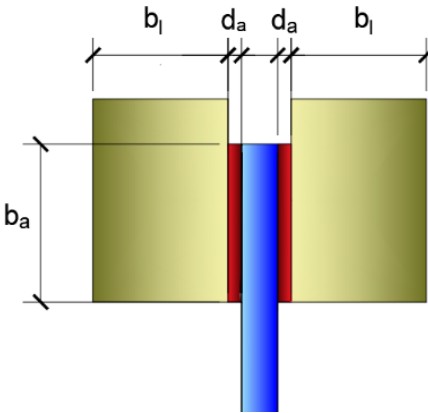

**Figure 4.** Geometry and usage symbols of the adhesive joint.

Consider the thickness of the adhesive joint:

$$\tau_{xy} = G \cdot \alpha = G \cdot \frac{\delta_{(x)}}{d_a} \tag{10}$$

Assuming equal displacements along the entire length, the force can be written as follows:

$$\int_0^1 dF_s \cdot dx = 2 \cdot G \cdot \frac{\delta}{d_a} \cdot b_a \cdot l \tag{11}$$

Using Moehler's hypotheses, we obtain:

$$\frac{K_{1/3}}{s_{1/3}} = \frac{2 \cdot G \cdot b_a}{d_a} \tag{12}$$

and

$$C_i \left( = \frac{K_i}{s_i} \right) = \frac{2 \cdot G \cdot b_a}{d_a} \tag{13}$$

We can now use the analogy from SIST EN 1995-1-1 [32],:

$$\gamma = \frac{1}{1+k} \tag{14}$$

and

$$k = \frac{\pi^2 \cdot E \cdot A}{L^2 \cdot K_k} \tag{15}$$

We write:

$$\gamma = \left(1 + \frac{\pi^2 \cdot E_i \cdot A_i \cdot s_i}{K_i \cdot l^2}\right)^{-1} = \left(1 + \frac{\pi^2 \cdot E_i \cdot A_i \cdot d_{a,i}}{2 \cdot G_a \cdot b_{a,i} \cdot l^2}\right)^{-1} \tag{16}$$

The shear modulus of the adhesive $G$ and the dimensions given in Figure 4, adhesive thickness $d_a$, adhesive height $h_a$, adhesive length $b_a$, determine the gamma factor ($\gamma_i$) of the adhesive or the stiffness of the adhesive.

The most important equation for determining the bending stiffness of such composite beams with different elastic moduli can be found in SIST EN 1995-1-1 [32], where the effective moment of inertia is given as follows:

$$I_{y,eff} = \sum_{n=1}^{3} n_i \cdot I_{yi} + \gamma_i \cdot n_i \cdot A_i \cdot a_i^2 \tag{17}$$

where:

$$n_i = \frac{E_i}{E_j} \tag{18}$$

$E_i$    modulus of elasticity of component $i$
$E_j$    modulus of elasticity of component $j$
$I_i$    moment of inertia of component i
$\gamma_i$    gamma value
$A_i$    cross—sectional area of component $i$
$a_i$    area of the $i$-th component ai distance between the local and global coordinate systems of the $i$-th component

Starting from our example (Figure 1), where $n_1 = n_3$, $\gamma_1 = \gamma_3$, $A_1 = A_3$ and $a_1 = a_3$, we write:

$$I_y = 2 \cdot I_{y1} + I_{y2} + 2 \cdot A_1 \cdot a_1^2 \cdot n_E^s \tag{19}$$

The effective moment of inertia of the cross section allows us to calculate the modulus of elasticity for timber in further calculations. Similar to the effective moment of inertia, we can derive an expression for the effective cross-sectional area. For our example (Figure 1), we can use Equation (17) to write:

$$I_{y,eff} = 4 \cdot \frac{b_l \cdot h_l^3}{12} + 4 \cdot b_b h_b \cdot \frac{H - h_b}{2}^2 \cdot \gamma + \frac{b_s \cdot h_s^3}{12} \cdot n_E^s \tag{20}$$

According to the design manual and considering the static model of the beam shown in Figure 5, we write:

$$M = \frac{F \cdot L}{3} \tag{21}$$

$$F = \frac{P}{2} \tag{22}$$

$$\mu = \frac{23}{648} \frac{F \cdot l^3}{EI} = 0,03549 \frac{Fl^3}{EI} \tag{23}$$

following:

$$K_H = \frac{EI}{0,03549 \cdot L^3} \tag{24}$$

$$K_H = \frac{2 \cdot G_a \cdot h_a}{d_a} \tag{25}$$

where:

$G_a$    shear modulus of the adhesive
$H_a$    the height of the adhesive application
$D_a$    adhesive thickness.

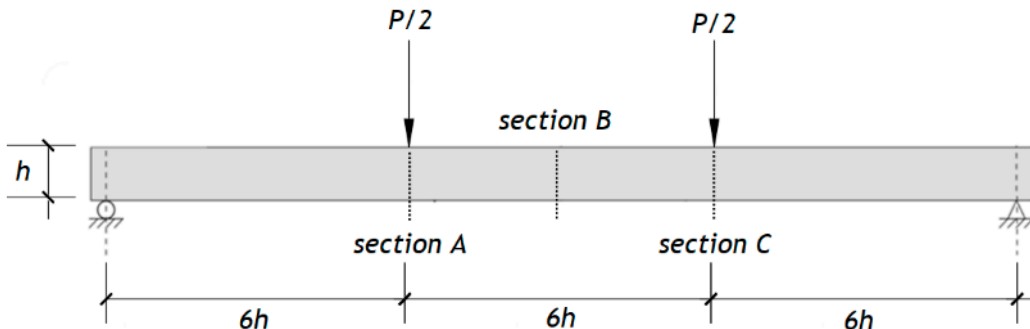

**Figure 5.** Setup for four-point bending.

## 3. Experimental Analysis

### 3.1. Static System

The bending tests for both specimens TGIB-fw and TGIB-gf were carried out according to the recommendations of EN 408 [33], which prescribe the performance of a four-point bending test, as shown schematically in Figure 5. The distance between the supports was 18 h (4320 mm) and the loading points were located in a third of the span.

The initial bending stiffnesses were obtained from the measured data just before the first crack appeared in the beam, and the final initial bending stiffnesses were obtained at the end at the maximum load. Both were calculated using the equations from Bernoulli–Euler beam theory

$$M_{max} = \frac{F_{max}}{2} \cdot a \tag{26}$$

where:

$M_{max}$   maximal bending moment
$F_{max}$   maximum load
$a$   distance between a loading position and the nearest support in a bending test

$$u_{cr} = \frac{F_{cr} \cdot a}{48 \cdot EI} \left( 3l^2 - 4a^2 \right) \tag{27}$$

where:

$u_{cr}$   vertical displacement at mid-span at the first crack in the beam
$F_{cr}$   load at the first crack in the beam
$EI$   bending stiffness of the beam
$l$   span of the specimen between the two supports

### 3.2. Cross—Section of TGIBs

#### 3.2.1. TGIB-fw

The research and manufacture of life-size TGIBs was carried out in the Materials and Structures Research Laboratory at University of Maribor, Faculty of Civil Engineering, Transportation Engineering and Architecture. The dimensions of the beams were carefully selected and discussed within the literature studies and the Wood Wisdom project to be as comparable as possible with the Swedish study (10), so that the dimensions were practically identical.

Basically, 18 test specimens were made from finger-jointed C24 timber with two different types of glass and three different adhesives. For comparison purposes, the test specimens were 4800 mm long and 240 mm high. They consisted of two types of 8 mm thick glass. Annealed float glass and fully tempered glass were used. The edges of both

types were polished. Timber flanges measuring 30 mm × 45 mm were glued to the glass web. The cross section is shown in Figure 6.

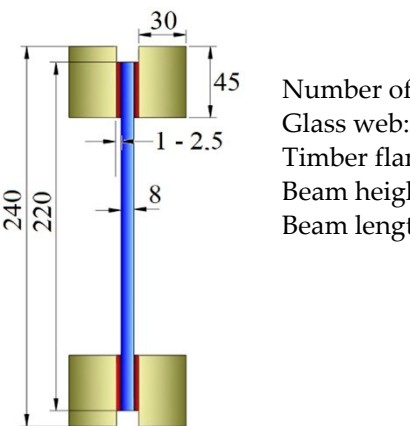

Number of specimens: 18
Glass web: 8/220—4800 mm
Timber flanges: 30/45 mm
Beam height: 240 mm
Beam length: 4800 mm

**Figure 6.** Cross—section of the beam specimens TGIB-gf.

### 3.2.2. TGIB-gf

Figure 7 shows a cross-section of a timber- glass I-beam, with a groove in the flange. This type of beam was used for comparison. All beams were 240 mm high and 4800 mm long. Two types of glass were used for the beams: annealed float glass (AF) and heat-strengthened glass (HS), 190 × 4800 mm with a thickness of 8 mm. The edges were polished. Finger-jointed pine studs 45 × 60 mm thick were used for the timber flanges. A 12 × 20 mm groove was cut in the flanges for mounting the glass pane. Thus, the thickness of the adhesive strip for all beams was 2 mm (on both sides of the glass strip). Three different types of adhesives with different stiffnesses were used for bonding: 3M DP490 (epoxy), SikaFast 5221 (acrylate) and Sikasil SG-500 (silicone).

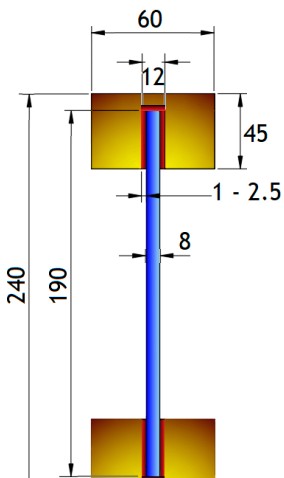

Number of specimens: 12
Glass web: 8/190—4800 mm
Timber flanges: 45/60 mm
Beam height: 240 mm
Beam length: 4800 mm

**Figure 7.** Cross section of the compared beams—TGIB-gf [10,18].

### *3.3. Material Properties*
### 3.3.1. Timber

Timber is a renewable and natural material with excellent mechanical properties and a very wide range of applications. It is used for load-bearing and non-load-bearing structural elements, for load-bearing and non-load-bearing cladding, and much more. It has always played an important role in the construction industry, but with the development of new knowledge about the mechanical and physical properties of timber and the development of new technologies, timber construction is also gaining importance in more architecturally

and structurally demanding areas. Timber in timber–glass composites has a low deformation capacity and usually shows brittle failure on the tensile side due to growth defects or finger joints. This is the most common cause of collapse of timber beams. Therefore, the obvious choice for composite construction is to use good quality timber that does not have defects—especially knots. For this composite, finger-jointed spruce and pine C 24 was used. Material properties of timber C24 are given in Table 2.

**Table 2.** Material properties of timber C24 [34].

| Property | Symbol with Units | Value |
|---|---|---|
| Density | $\rho$ [kg/m$^3$] | 420 |
| Bending strength | fm [N/mm$^2$] | 24 |
| Compressive strength | fc [N/mm$^2$] | 25 |
| Modulus of elasticity | E [N/mm$^2$] | 11,000 |

3.3.2. Glass

Today, soda–lime–silica glass is mainly used in the construction industry. Table 3 shows general physical properties. In general, we must consider the fact that glass has no built-in safety mechanism. It deforms to its elastic limit or breaks brittle. Depending on the failure mechanisms of structural glass, we distinguish: float glass, annealed float glass, heat-strengthened glass, fully tempered glass, chemically strengthened glass and laminated glass.

**Table 3.** General physical properties of soda-lime-silica glass [35].

| Property | Symbol with Units | Value |
|---|---|---|
| Transition temperature | $T_g$ [°C] | 564 |
| Liquid temperature | $T_l$ [°C] | 1000 |
| Density | $\rho$ [kg/m$^3$] | 2500 |
| Coefficient of thermal expansion | $\alpha_T$ [K$^{-1}$] | $0.9 \times 10^{-5}$ |
| Thermal conductivity | $\lambda$ [W/(m K] | 1.0 |
| Specific heat capacity | c [J/(kg K] | 720 |

Three types of structural glass were used for comparison in this study—annealed float glass (AF), fully tempered glass (FT) and heat-strengthened glass (HS).

Annealed float glass is essentially float glass made by a process in which the float glass is cooled slowly enough to avoid internal stresses. Glass can be made more load-resistant by inducing the compressive stresses on the surface. It is annealed when it is heated beyond the transition point and then allowed to cool slowly [36]. All annealed float glass becomes brittle during the production process. The tensile bending strength of tempered glass is 45 N/mm$^2$ (Table 4). Such glass is very brittle and breaks into large pieces that can cause serious injury and damage. Such glass should not be used for large glazing or anywhere where there is a greater risk of injury to people if it breaks.

**Table 4.** Material properties of used structural glasses [35].

| Material | Modulus of Elasticity E [N/mm$^2$] | Tensile Bending Strength $f_{mt}$ [N/mm$^2$] |
|---|---|---|
| AF glass | 70,000 | 45 |
| HS glass | 70,000 | 70 |
| FT glass | 70,000 | 120 |

Fully tempered glass (FT) is made from annealed glass by a thermal tempering process. Unlike annealed float glass, this type of glass breaks into small pieces and reduces the risk of breakage. The tensile bending strength of tempered glass is 120 N/mm$^2$ (Table 4). After tempering, the temperature resistance, impact strength, and bending strength increase. FT glass must be cut to the correct size and pressed into shape before tempering, as it cannot

be reworked after tempering. Therefore, no post-critical strength is observed here. The first crack corresponds to the final failure.

Heat-strengthened glass is the most common type of toughened glass used in safety components. It is less than 12 mm thick and is tempered to produce residual stresses at the surface, but at a lower temperature and cooling rate than fully toughened glass. It has lower residual stresses and breaks into what appear to be larger pieces, but they are still smaller than annealed glass. Its tensile bending strength is about 70 N/mm$^2$ (Table 4).

### 3.3.3. Adhesive

Combining materials as diverse as timber and glass presents a unique challenge. The joint must strike the right balance between load-bearing capacity and ductility. In addition to the right choice of adhesive, the type of joint and the preparation and care of the surface are also important. When choosing an adhesive, it is important to know the criteria that each adhesive meets. These criteria are the strength of the adhesive bond, the deformability of the adhesive, the exposure to moisture and climatic changes, the temperature (in)resistance, the degree of adhesion, the type and bonding time of the adhesive, the viscosity, the humidity, and the fire resistance of the adhesive. One of the main advantages of a bonded joint is that it achieves uniform stress along the entire length of the joint, unlike the high local stresses of nailed or joints. Cruz and Pequeno [5] were among the first to investigate adhesives for joining timber–glass composites. They divided them into three groups, namely:

- rigid adhesives—extremely high strength and stiffness (e.g., acrylic acrylate, epoxy);
- medium stiff adhesives—balanced strength and elasticity (e.g., polyurethane);
- elastic adhesives—extreme elasticity and low strength (e.g., silicone and some polyurethanes).

Combining two materials with significantly different values of coefficient of thermal expansion ($\alpha_T$), an elastic adhesive can be used to mitigate the relative deformation between the two materials due to temperature changes. The elastic adhesive bond performs a similar function in the event of a sudden change in relative humidity that affects the volumetric properties of the timber (i.e., expansion and contraction). Three types of adhesives were used: SikaSil SG500 silicone, SikaFast 5215 acrylate and Sikadur®-31 CF epoxy (Table 5).

**Table 5.** Material properties of used adhesives.

| Adhesive | Modulus of Elasticity E [N/mm$^2$] | Shear Modulus G [N/mm$^2$] |
|---|---|---|
| Silicone: SikaSil SG500 | 1.1 | 1 |
| Acrylate: SikaFast 5215 | 78 | 26.73 |
| Epoxy: Sikadur®-31 CF | 4000 | 1406.5 |
| Epoxy: 3M DP490 | 1595 | 560.83 |

Figure 8 shows diagrams demonstrating the significantly higher strength of epoxy compared to silicone. The epoxy adhesive is shown to have elastic, nearly linear behaviour until failure. In addition, the stresses in epoxy adhesives can be up to 100 times lower than in silicone, which should be considered when selecting the type of adhesive. Acrylate adhesives exhibit bilinear behaviour.

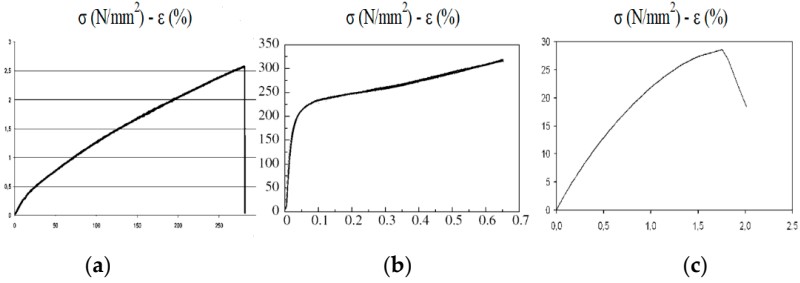

**Figure 8.** σ-ε diagrams of different types of adhesives in tension; (**a**) silicone [1], (**b**) acrylate [18], (**c**) epoxy [1].

In practice, we are usually dealing with medium- to long-term loads. Therefore, we should refer to the results of Haldimann et al. [36], who showed that the long-term strength of silicone is only about 10% of its short-term strength due to the highly creeping behaviour of silicone sealants. It is also worth mentioning the findings of Cruz et al. [37], based on the results of shear tests, showing that the failure mode of timber and glass elements depends on the strength of the adhesive. In general, it can be concluded that glass regularly fails in combination with highly resistant adhesives.

### 3.4. Manufacturing of TGIBs

#### 3.4.1. TGIB-fw

#### Silicone

Sika Slovenia has recommended the use of Sikasil SG 500 silicone. This is a two-component, highly modular structural silicone adhesive manufactured on a neutral base. It is mainly used for structural bonding of glass and other demanding industrial applications. It is applied with a compressed air gun, as it begins to cure immediately after mixing the two components. The speed of the reaction itself depends mainly on temperature [38].

The surface treatment depends on the type and behaviour of the surface and is crucial for a durable bond. Preparation for bonding requires patience and time. Following the manufacturer's recommendations, we first used Sika ADPrep cleaner (Figure 9a) and clamped the timber flanges to the surface at an appropriate distance using coupling pieces (Figure 9b). The timber webs were then cleaned and the clean glass plate was placed on top. Double-sided tape was applied to the glass plate in advance to ensure that the adhesive did not exceed the intended application width of exactly 35 mm. In addition, 2.5 mm spacers were placed between the timber and the glass to maintain the thickness of the adhesive across the entire substrate (Figure 9c).

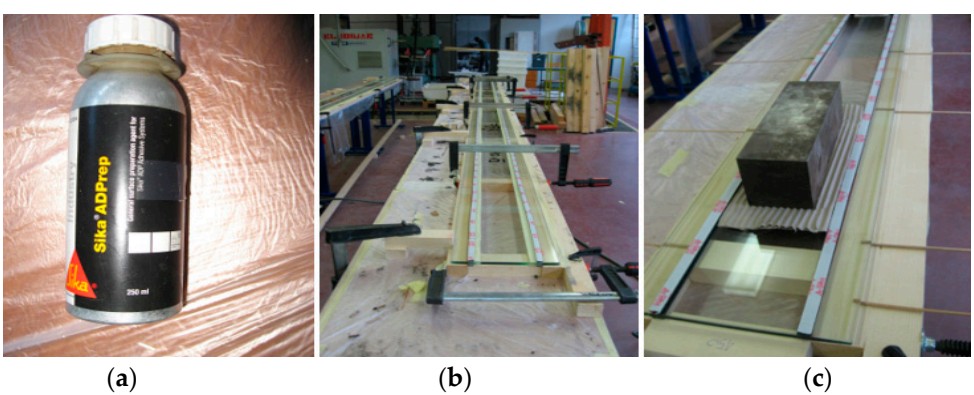

(a)　　　　　　　　　　　(b)　　　　　　　　　　　(c)

**Figure 9.** (**a**) Sika ADPrep Cleaner; (**b**) preparation of both materials for bonding by setting the prescribed distances; (**c**) use of spacers to maintain the intended adhesive thickness.

The mixing ratio for the Sikasil®SG-500 silicone used is A: B 10:1. The two components should be mixed homogeneously and without air bubbles in the correct ratio, as indicated, with an accuracy of $\pm$10%. The optimum temperature for the substrate and the sealant is between 15 °C and 25 °C. Component B is sensitive to moisture and should therefore only be exposed to air for short periods. For this purpose, we used a compressed air gun from SikaSlovenija, who also provided us with expert assistance during the bonding process [38]. The final appearance of the TGIB can be seen in Figure 10c.

#### Acrylate

The acrylate SikaFast 5215 NT is a two-component adhesive for structural bonding. It is fast curing, which requires special attention from the user. The open bonding time is influenced by the ambient temperature. The higher the temperature, the shorter the bonding time. The adhesive has high strength and good impact resistance. It is partially elastic and odourless. It is acrylic-based and contains no solvents or acids. It can be used to

bond a wide variety of materials, including metals and plastics in addition to timber and glass [39].

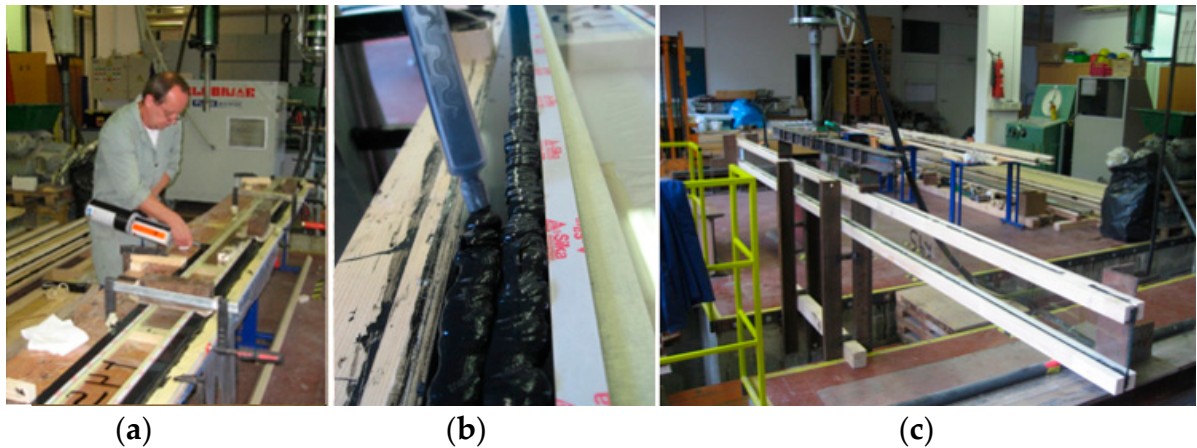

(a)  (b)  (c)

**Figure 10.** (**a**) application of the silicone between the glass and the timber; (**b**) application of the silicone on the top of the glass; (**c**) final appearance of the timber and glass I-beam.

Again, both materials had to be cleaned first. We used Sika Cleaner 205. The adhesive itself was applied using a compressed air gun (Figure 11a), which allows for appropriate dosing and mixing of components A and B. The mixing ratio A:B for SikaFast®-5215 NT was 10:1 (±10%). The mixed adhesive has an open time of about 5 min and reaches its working strength (curing time) in about 15 min. The optimum temperature for the bonding process is between 15 °C and 25 °C. The influence of temperature on reactivity must be considered. After the open time, the bonded parts can no longer be adjusted. In this case we were supported by the experts from SikaSlovenija, who provided expert assistance during the bonding process [39].

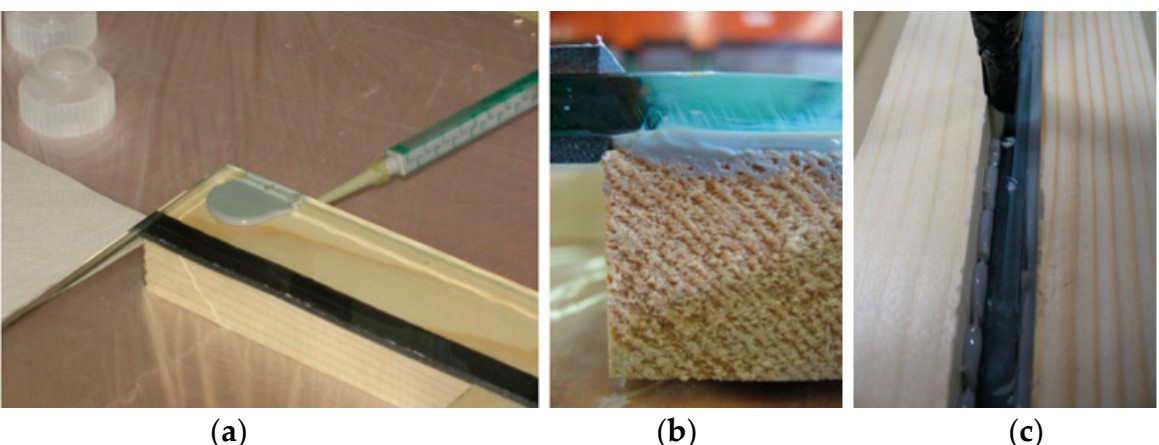

(a)  (b)  (c)

**Figure 11.** (**a**) application of the acrylate between the glass and the timber; (**b**) the appearance after the first bonding; (**c**) the application of the adhesive between the second flanges and the glass web.

The application thickness is limited to a minimum of 0.5 mm and a maximum of 3 mm. In our case, we used spacers so that the thickness was 2.5 mm, and the width was 35 mm. Bonding was similar to silicone, where we first bonded the lower, fixed part of the timber flanges and the glass. Due to the extremely fast bonding time, the bonding process for the upper timber flanges was adjusted to fill the space in the vertical position (Figure 11c), which was a disadvantage as we could not verify that the space had been filled [39].

After gluing, the beams had to rest for at least 24 h. Then all joints were removed (Figure 12a). The final appearance of the composite can be seen in Figure 12b.

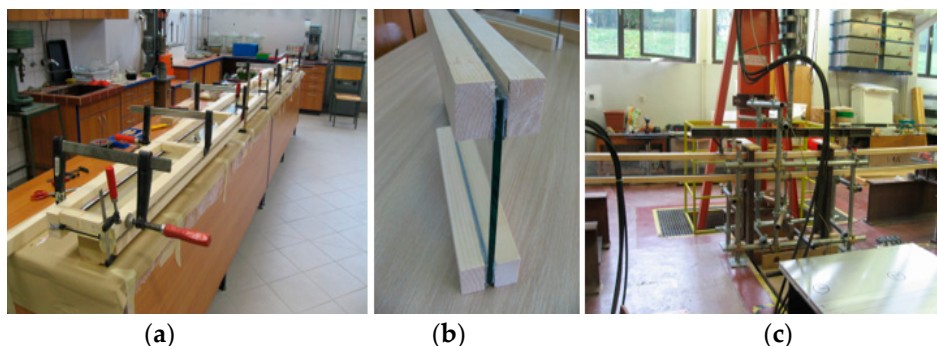

**Figure 12.** (**a**) glued beam; (**b**) final appearance of TGIB with acrylate adhesive; (**c**) TGIB during four-point bending test.

Epoxy

Sikadur®-31 CF Normal is a two-component, thixotropic adhesive based on epoxy resins and special fillers. Insensitive to moisture, it can be used in a temperature range between +10 °C and +30°C. It is easy to mix and install, adheres very well to most building materials, and has a high bond strength. It cures without shrinkage, has high initial and breaking strength, has good chemical and abrasion resistance, and is impermeable to liquids and water vapour [40]. The two mixing components, which are kept in a metal container (Figure 13a), are coloured differently to facilitate mixing control. The mixing ratio is prescribed as component A (white) to component B (grey) = 2:1, by weight or volume. The working time starts at the moment the resin and hardener are mixed. It is shorter at high temperatures and longer at low temperatures. The larger the volume of the mixture, the shorter the bonding, and thus, the processing time. To prolong the workability at high temperatures, it is necessary to divide the mixed adhesive into smaller parts or to cool the A + B components before mixing. An illustration of the mixture is shown in Figure 13b, which was applied to the glass with a spatula (Figure 13c).

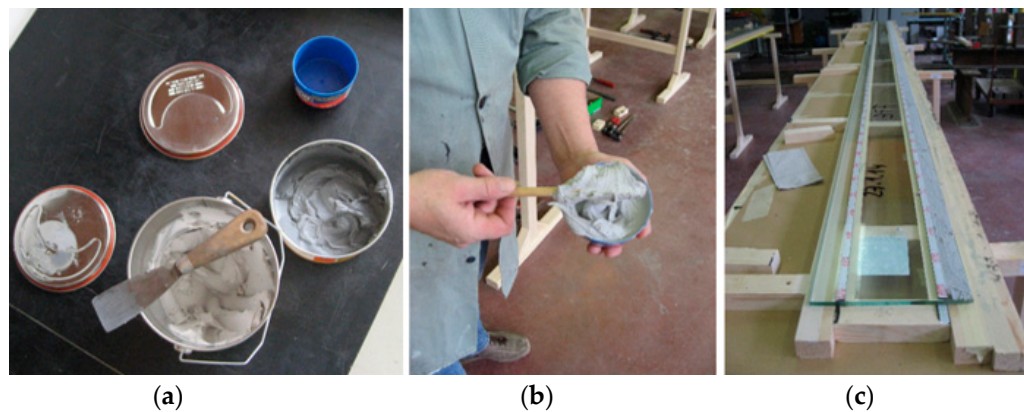

**Figure 13.** (**a**) Epoxy Sikadur®-31 CF Normal component A—white and component B—grey; (**b**) shows the mixture of the two components of the epoxy adhesive; (**c**) application of the epoxy adhesive to the glass.

Again, both the timber and the glass must be cleaned as the surface is required to be clean, dry, or damp (no standing water) and free from dirt, grease, oils, and old layers and coatings. The thickness of the layer is also prescribed at a maximum of 3 mm. The thickness prescribed for our gluing of the beams was 1 mm and 35 mm wide. This was ensured with the help of wire spacers (Figure 14a). After bonding, the prepared TGIB was weighted down and left for 4–7 days (Figure 14b). Figure 14c shows this beam during the bending test.

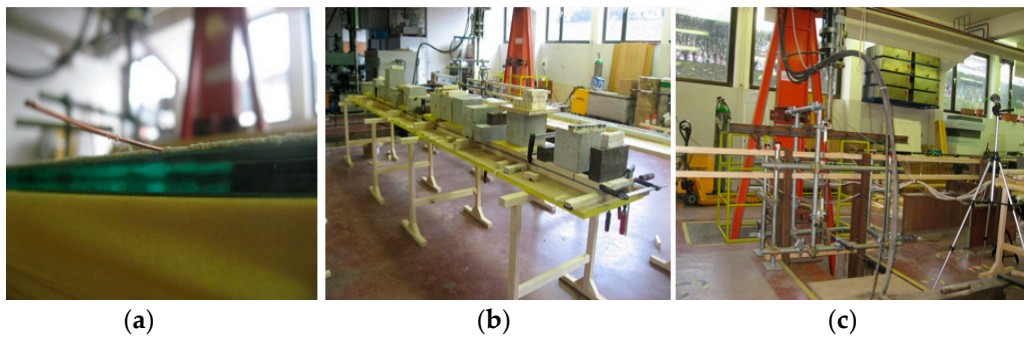

|            |            |            |
|:----------:|:----------:|:----------:|
| (**a**)    | (**b**)    | (**c**)    |

**Figure 14.** (**a**) wire spacer to secure the correct thickness of epoxy adhesive; (**b**) weighting of the already bonded TGIB; (**c**) TGIB with epoxy adhesive during testing.

### 3.4.2. TGIB-fw

The preparation of TGIB samples with grooved flanges, as described in Kozlowski et al. is much simpler. First, the glass plate was cleaned with alcohol. Masking tape was used to protect the glass surface from the adhesive. All adhesives were supplied in pre-packaged containers and applied with static mixers. Before applying the adhesive, the flange was protected with a masking film, as was the tape. The adhesive was poured into the groove with a compressed air gun and the wet adhesive was spread with a spatula. The flange was then placed under the tape, which was then lowered and placed in the groove and stabilised. At this stage accuracy was critical, so the height of the half beam was measured very accurately before the tape was applied to the side beams. Rubber strips were used to ensure even bonding of the adhesive line thickness. The excess wet adhesive was then removed with a putty knife so that the masking tape could be removed. Once the tape was in place in the mounts, it was impossible to remove the rubber strips without tearing them off, so they were cut short and left in the groove. The stabilised specimens were left until the adhesive had cured.

### 3.5. Measuring Position and Experimental Test
### 3.5.1. TGIB-fw

For the first of 18 specimens, strains at glass webs and timber flanges, and vertical and relative displacements between glass and timber were measured simultaneously. Strains were measured in two sections located in close proximity to the loading points (sections A and C in Figure 8). On the glass track, the strain measurement points were positioned at the top and bottom edges where the highest strain values were expected. The strain measurement points on the timber were distributed on the bottom of the top flange and on the bottom and top of the bottom flange. All strain measurement points were equipped with strain gages (SGs) from HBM (for measurements on glass, the length of the SG measurement grids was 6 mm and for strain measurements on timber, SGs with 100 mm long measurement grids were used). All strain gages on the specimen were connected as a quarter Wheatstone bridge with four wires [41]. All vertical displacements at mid-span were measured using inductive displacement transducers. They were also used to measure relative displacements between glass and wood in the horizontal direction at one end of the composite beam. Measurements were made using the HBM MGCPlus data acquisition system controlled by a laptop running Catman 4.5 licensed software. The strains of the glass and wood were also measured at mid-span of the subsequent specimens (section B in Figure 8). The intensity of strain was measured directly on the hydraulic piston using a set of four strain gages connected in a full Wheatstone bridge. Analytical and experimental calibration was performed to determine the strain intensity from the measured strains.

A hydraulic piston was used for loading (Figure 15a). The load was applied symmetrically via a weight yoke (Figure 15b), which allowed us to transfer the force via the hydraulic piston to the TGIBs as a two-point load ($P/2$).

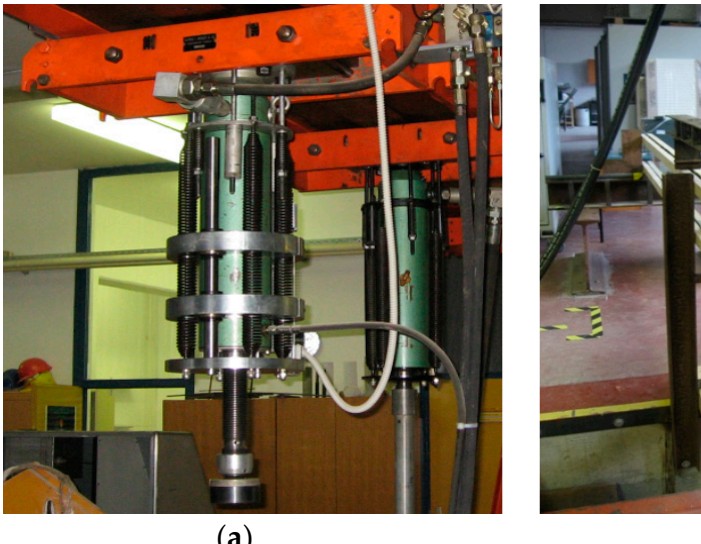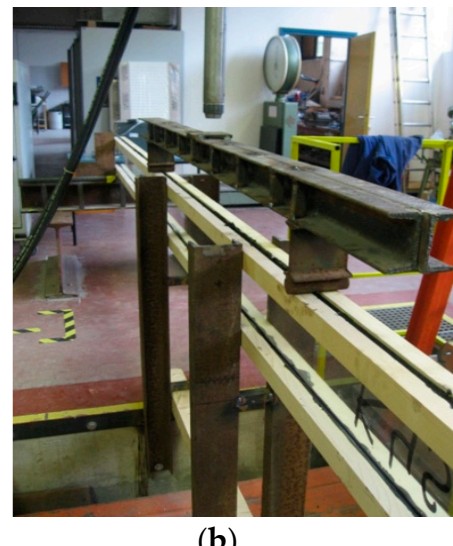

(**a**)　　　　　　　　　　　　　　　(**b**)

**Figure 15.** (**a**) hydraulic piston; (**b**) weight yoke through which the force of the piston was transmitted to the specimen.

Steel plates (Figure 16) were placed under the yoke to prevent the yoke from digging into the timber during the test.

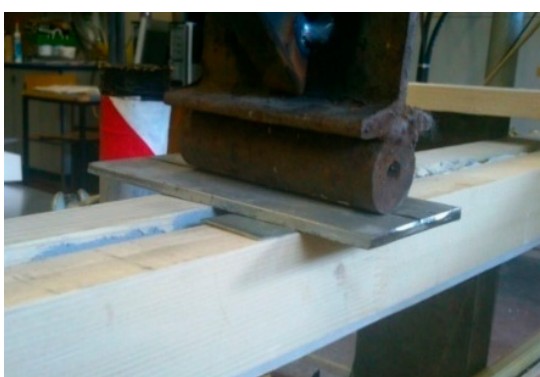

**Figure 16.** Steel plates to prevent the weight yoke from pushing into the timber.

The specimens were placed in prepared steel frames (Figure 17) that prevented the beams from slipping. They were loaded with a hydraulic piston with a force of 1 kN/100 s until final failure. The specimen during loading is shown in Figure 18, where the first cracks in the glass are already visible.

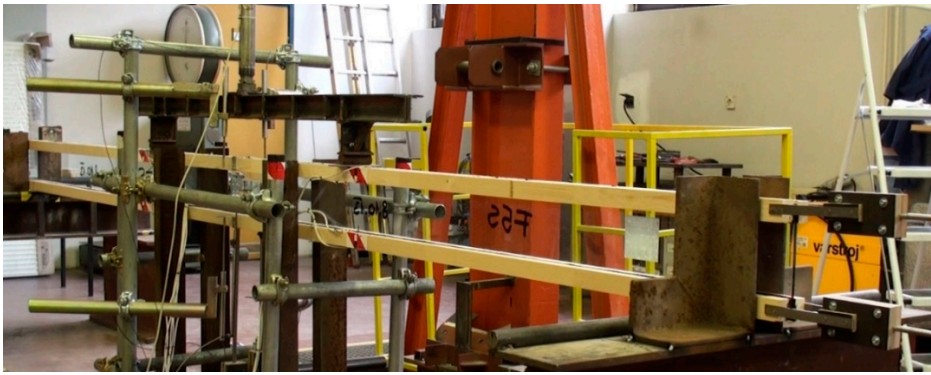

**Figure 17.** Positioning the specimen between the steel profiles and attaching the weight yoke.

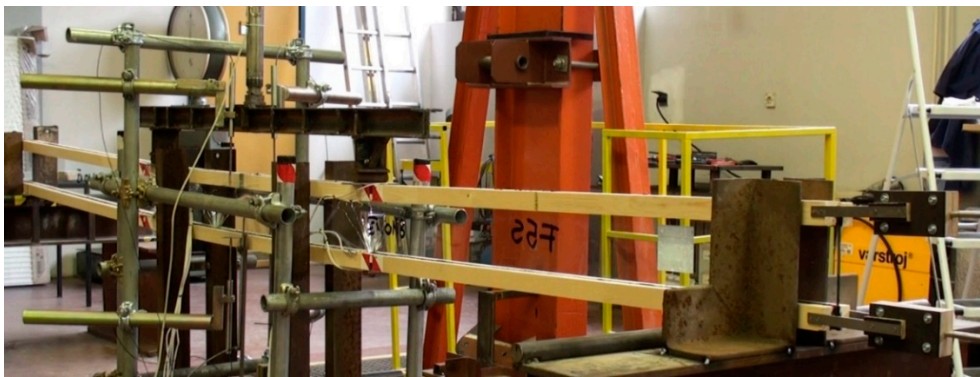

**Figure 18.** TGIB during four-point bending test.

### 3.5.2. TGIB-gf

All TGIB-gf specimens were prepared also by hand. The glass plate was first cleaned with alcohol. The flange and the web were protected with a masking film before applying the adhesive. The adhesive was applied into the groove using a compressed air gun and a spatula. The flange was placed under the glass web, which was then positioned and stabilised in the groove. Rubber strips were used to ensure even uniform bonding of the adhesive line thickness. Once the tape was placed in the rubber strips were shortened and left in the groove. The stabilised specimens remained until the adhesive had cured. Unlike acrylate, which took approximately 10 min to cure, the specimens were bonded with silicone and epoxy adhesive for 12 h before the other flange was bonded to the glass web.

### 4. Results

All tests were performed in the laboratory under constant conditions of temperature and humidity. The room temperature and humidity data, as well as the moisture content of the wood, were measured during the preparation of the samples and during the tests. Temperature and humidity have a great influence on the material properties of wood. Table 6 contains the measurements of relative humidity and air temperature during the preparation of the specimens (P) and during the testing of specimens (T), as well as the measured mean values of relative humidity on the wood according to each test group.

**Table 6.** Environmental condition during the preparing the specimens (P) and during the testing of specimens (T) and moisture of wood during the preparing the specimens and during the testing of specimens.

|  | Air | | Specimen |
|  | Temperature [°C] | Humidity [%] | Humidity [%] |
|---|---|---|---|
| Mean value S_AF (P) | 21.3 | 61.5 | 12.6 |
| Mean value S_AF (T) | 21.5 | 61.3 | 11.3 |
| Mean value A_AF (P) | 21.3 | 61.2 | 11.8 |
| Mean value A_AF (T) | 21.6 | 61.3 | 11.4 |
| Mean value E_AF (P) | 21.8 | 60.8 | 12.5 |
| Mean value E_AF (T) | 22 | 60.1 | 11.4 |
| Mean value S_FT (P) | 21.5 | 61.2 | 11.8 |
| Mean value S_FT (T) | 21.4 | 61.3 | 11.3 |
| Mean value A_FT (P) | 21.4 | 60.8 | 12.7 |
| Mean value A_FT (T) | 21.3 | 61.0 | 11.7 |
| Mean value E_FT (P) | 21.6 | 60.5 | 12.2 |
| Mean value E_FT (T) | 21.8 | 60.7 | 11.3 |

### 4.1. Experimental Analysis

For the experimental analysis, we can now summarise: 18 TGIB-fw test specimens were made, nine with annealed float glass and nine with fully tempered glass. The group

with annealed float glass (AF) was bonded with three adhesives. Three specimens were bonded with silicone SikaSil SG 500, three with acrylate SikaFast 5215 and three with epoxy. The group of fully tempered glasses (FT) was also bonded with the same three adhesives, i.e., three specimens with silicone SikaSil SG 500, three with acrylate Sika Fast 5215 and three with epoxy Sikadur®-31 CF Normal.

For TGIB-gf, 12 specimens were prepared, six with annealed float glass and six with heat-strengthened glass. The group with annealed float glass (AF) was bonded with only one adhesive, 3M DP490 epoxy. The heat-strengthened glass test group was bonded with three adhesives, i.e., two samples with SikaSil SG 500 silicone, two with Sika Fast 5215 acrylate and two with 3M DP490 epoxy.

Both test groups, TGIB-fw and TGIB-gf, were tested in a four-point bending test according to the recommendations of EN 408. For the comparison between the cross sections of TGIB-fw and TGIB-gf, only the specimens bonded with epoxy adhesive were considered for the annealed float glass specimens. For TGIB-fw, there were no other comparable specimens. However, for fully tempered glass and heat-strengthened glass, we compared the specimens with respect to all three adhesives used—silicone, acrylate, and epoxy.

### 4.1.1. TGIB-fw

Figure 19 shows the force versus absolute value of average vertical beam displacement for TGIB with annealed float glass. A multistage failure mechanism is observed here. In stage I, the relationship between force and displacement is almost perfectly linear elastic until the first crack in glass occurs. Thereafter, there is a sudden drop in bending stiffness and an increase in vertical displacement. After the first crack in the glass, which occurs directly under the load point, the bottom edge of the timber flanges acts as a bridge between the cracks, which, together with the uncracked zone at the middle and top edges, ensures that the beam can continue to carry the load. In the next phase, the existing crack is enlarged and a new crack forms below the second load point. As the process continues, more and more cracks form, usually between the two load points. The tensile forces are thus transferred from the glass web to the timber flange. This distribution of forces prevents brittle collapse and ensures the ductility of the whole system. This is called postcritical strength [21], which thus defines the increased value of the load from the first crack in the glass to the final failure.

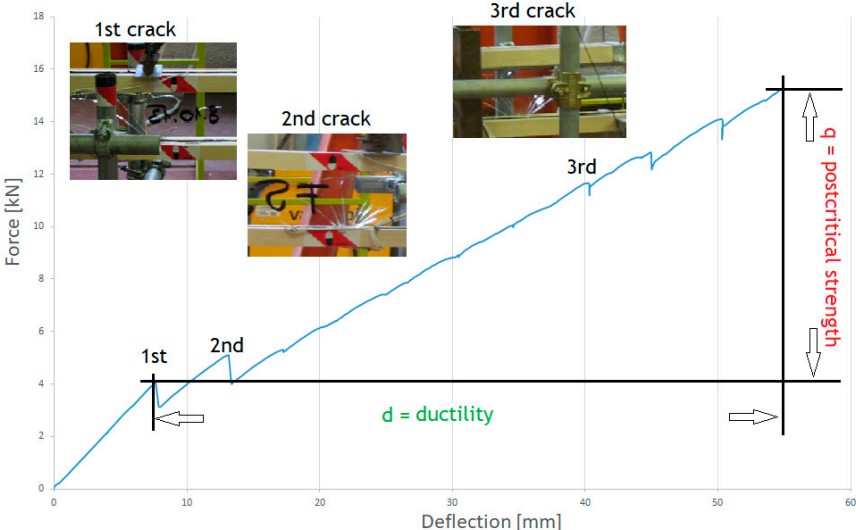

**Figure 19.** Force versus vertical beam displacement for TGIB with annealed float glass.

The results for annealed float glass from all three test groups, summarised in Figure 20, show that the failure pattern is the same for all three adhesives, with a multistage failure mechanism occurring in all test specimens. The bending stiffness of the TGIB-fw is constant

until the first crack occurs. Thus, the theory described above that the beam continues to carry the load until final failure despite cracks in the glass is confirmed here by a practical example. The first crack implies a reduction in bending stiffness, an increase in vertical displacements and a transfer of tensile stresses through the adhesive to the lower wooden flange, which contributes to an increased ductility of the beam. The crack formed continues to grow and a new crack form on the other side where the force was applied. Later, more cracks form along the entire length of the beam. Specimens with stiffer adhesives (acrylate and epoxy) have higher bending stiffness. The first cracks appear earlier in silicone specimens than in acrylate or epoxy specimens, which is due to the stiffness of the adhesives, silicone being the most elastic in this case. The graph shows that for epoxy the first cracks occur at a force of about 10 kN, for acrylate it is about 6 kN and for silicone these forces are lower, about 4 kN. The exact results can be found below in Table 3. The decrease in stiffness is also most noticeable and visible with the most elastic adhesive—silicone.

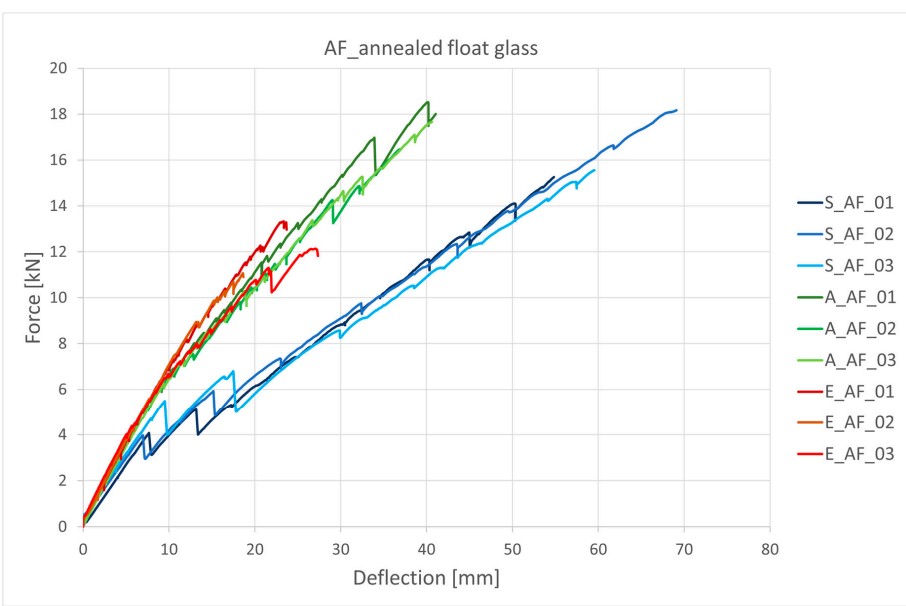

**Figure 20.** Diagram force vs. deflection for the annealed float glass (adhesive: S—silicone Sika Sil SG500, A—acrylate Sika Fast 5215, E—epoxy Sikadur®-31 CF Normal, AF—annealed float glass).

Table 7 shows the numerical values recorded during the test. For silicone, the first crack ($F_{cr}$) occurs on average at a force of 4.7 kN, while for acrylate, a 1.2 kN higher average force value was recorded at the first crack compared to silicone. For epoxy, this force was measured at 10.6 kN. The average maximum force for the acrylate specimens was 17.6 kN, followed by silicone at 16.4 kN. The lowest values were recorded for the epoxy adhesive specimens at 12.1 kN. The highest deviation was recorded for the silicone bonded test specimens. For epoxy adhesives, these displacements were on average three times smaller. From the obtained results, it can be concluded that $F_{cr}$ strongly depends on the stiffness and on the thickness of the adhesive and the displacements also depend on the stiffness of the adhesives. The results of load increases from first crack to failure also show similar results. Silicone generally provides higher ductility than epoxy or acrylate.

The results in Table 7 show that the maximum displacements ($u_{max}$) are expected for the most elastic adhesive, i.e., silicone, with an average value of 61.1 mm, followed by the acrylate specimens with an average value of 39.5 mm and finally the specimens with the most rigid adhesive epoxy, with an average value of 23.2 mm. As expected, the ductility values, given by the ratio of the maximum force ($F_{max}$) to the force at the first crack ($F_{cr}$), were highest for silicone at 5.3, and lowest for epoxy at 1.3. The bending stiffness of the specimens are then inversely proportional to the ductility. The highest values were found for the test specimens bonded with epoxy adhesive, followed by acrylate and finally silicone.

**Table 7.** Results of measurements for TGIB-fw with annealed float glass (AF).

| Material | $F_{cr}$ [kN] | $F_{max}$ [kN] | $u_{cr}$ [mm] | $u_{max}$ [mm] | Ductility d = $u_{max}/u_{cr}$ | q = $F_{max}/F_{cr}$ | EI [MNm²] |
|---|---|---|---|---|---|---|---|
| S_AF_01 | 4.6 | 15.3 | 11.8 | 54.8. | 4.7 | 3.3 | 0.714 |
| S_AF_02 | 4.1 | 18.2 | 9.9 | 69.1 | 7.0 | 4.4 | 0.753 |
| S_AF_03 | 5.3 | 15.6 | 13.0 | 59.5 | 4.6 | 2.9 | 0.746 |
| **Mean value** | **4.7** | **16.4** | **11.6** | **61.1** | **5.3** | **3.6** | **0.738** |
| A_AF_01 | 6.1 | 18.5 | 9.6 | 41.1 | 4.3 | 3.0 | 1.155 |
| A_AF_02 | 5.3 | 16.5 | 7.8 | 36.9 | 4.7 | 3.1 | 1.237 |
| A_AF_03 | 6.2 | 17.7 | 9.9 | 40.6 | 4.1 | 2.9 | 1.138 |
| **Mean value** | **5.9** | **17.6** | **9.1** | **39.5** | **4.3** | **3.0** | **1.176** |
| E_AF_01 | 12.1 | 13.1 | 21.3 | 23.7 | 1.2 | 1.1 | 1.088 |
| E_AF_02 | 8.9 | 11.1 | 14.2 | 18.7 | 1.4 | 1.3 | 1.230 |
| E_AF_03 | 10.7 | 12 | 21.6 | 27.3 | 1.4 | 1.1 | 0.971 |
| **Mean value** | **10.6** | **12.1** | **19.0** | **23.2** | **1.3** | **1.2** | **1.096** |

Fully tempered glass (FT) is safer and has up to three times the bending strength of annealed float glass. Figure 21 shows the force (F) and displacement (u) diagram for TGIB-fw with FT glass. Practically all samples showed the expected linear behaviour. Only sample S_ FT _03 showed a deviation from the linear behaviour. This may be mainly due to the design of the test, as in this case no steel plate was inserted between the weight yoke and the timber beam to prevent the yoke from pressing into the timber. The second reason is due to the material properties of silicone. Silicone is extremely elastic and, in this case, only the adhesive can be plasticized. No special observations were made with the acrylic specimens. In the samples bonded with epoxy adhesives, a defect occurred in sample E_ FT _02. Here, collapsing of the adhesive was observed, causing the weight yoke to line the glass web and we had to stop the test. This was mainly due to the stiffness of the epoxy adhesive and the thickness of the bonded line, which was 1 mm. Therefore, the last measured force of 20.7 kN corresponds to the actual failure.

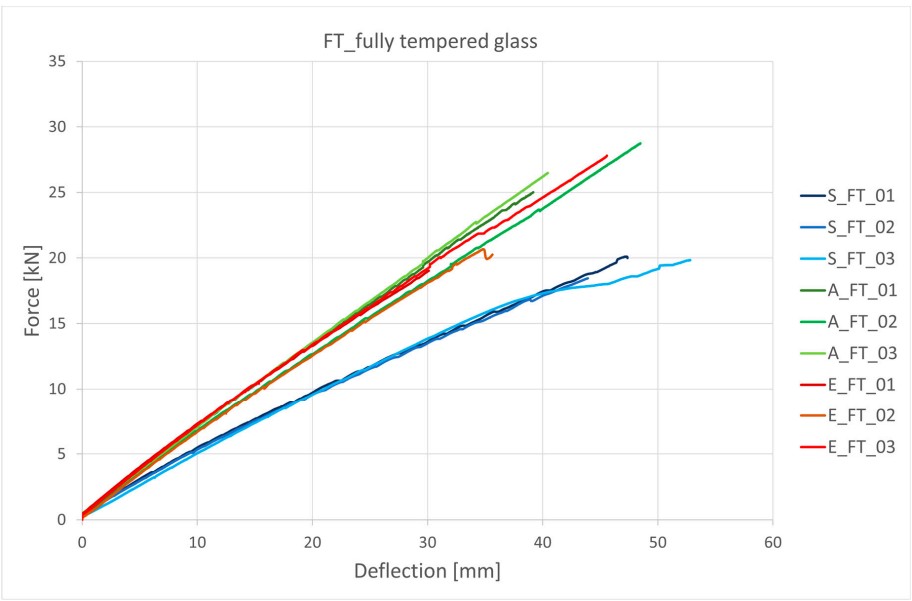

**Figure 21.** Diagram force vs. deflection for the fully tempered glass (adhesive: S—silicone Sika Sil SG500, A—acrylate Sika Fast 5215, E—epoxy Sikadur®-31 CF Normal, FT—Fully.

In fully tempered glass no post-critical strength is observed here. The first crack corresponds to the final failure, as shown in Table 8. This value, $F_{max}$, can be up to twice the

value for the annealed float glass sample. $F_{crack}$ and $F_{max}$ are therefore the same. Therefore, the parameter $q = 1.0$ and the ductility $d = 1.0$ for all test samples and are thus not specially presented in Table 7.

**Table 8.** Results of measurements TGIB-fw for fully tempered glass (FT) with q = 1.0 and d = 1.0.

| Specimen | $F_{cr} = F_{max}$ [kN] | $u_{cr} = u_{max}$ [mm] | EI [MNm2] |
|---|---|---|---|
| S_FT_01 | 20.0 | 47.4 | 0.604 |
| S_FT_02 | 18.5 | 43.9 | 0.603 |
| S_FT_03 | 19.8 | 52.8 | 0.537 |
| **Mean value** | **19.4** | **48.0** | **0.594** |
| A_FT_01 | 24.9 | 39.2 | 0.909 |
| A_FT_02 | 28.8 | 48.5 | 0.850 |
| A_FT_03 | 26.5 | 40.5 | 0.936 |
| **Mean value** | **26.7** | **42.7** | **0.898** |
| E_FT_01 | 19.0 | 30.1 | 0.905 |
| E_FT_02* | / | / | / |
| E_FT_03 | 27.8 | 45.6 | 0.872 |
| **Mean value** | **23.4** | **37.9** | **0.889** |

The highest load was observed in the acrylate specimens, where the average value was 26.7 kN and the highest measured value was as high as 28.8 kN. In the case of the epoxy adhesive, there is considerable variation in the maximum load values. It should be noted that the specimen E_ FT _02 * was not taken into account in the calculation of the mean values for the maximum load and displacement, since the failure occurred after the failure of the adhesive. The experimental analysis data shows that E_ FT _03 has a maximum load 8.8 kN higher than E_ FT _01. The answer here can be sought mainly in the precision of the bonding of the timber-glass joint. The epoxy adhesive here was provided at a thickness of 1 mm, was applied with a spatula, and there may have been an error here which was consequently reflected in the results. The silicone showed the lowest mean values of maximum load and the highest mean values of displacement, which was expected due to its elasticity. On average, the displacements of the silicone specimens were 5.3 mm higher than those of the acrylate and 10.13 mm higher than those of the epoxy adhesive.

### 4.1.2. TGIB-gf

Figure 22 shows the mid-span force-displacement curves for TGIBs with annealed float glass and heat-strengthened float glass [18].

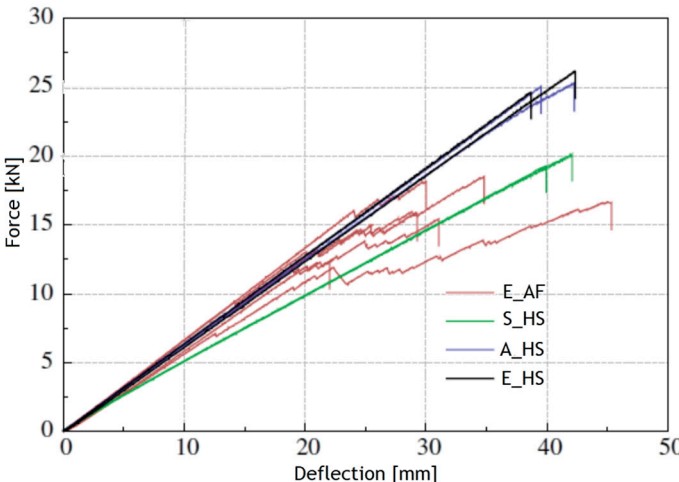

**Figure 22.** Beam tests TGIB-fg [18] (E—epoxy (3M DP490), S—silicone (Sika Sil SG500), A—acrylate (Sika Fast 5215) adhesive; AF annealed, HS heat strengthened float glass).

In Table 9 results of measurements of TGIB-gw with annealed float glass are given. These experimental studies of TGIBs-gf show results from specimens with epoxy adhesives. They show similar behaviour to TGIBs-fw with annealed float glass, i.e., the fracture values are much higher than the appearance of the first cracks in the glass.

**Table 9.** Results of measurements and of bending stiffnesses of TGIB-gw with annealed float glass—AF (E—epoxy, S—silicone, A—acrylate).

| Specimen | $F_{cr}$ [kN] | $F_{max}$ [kN] | $u_{cr}$ [mm] | $u_{max}$ [mm] | Ductility d = $u_{max}/u_{cr}$ | q = $F_{max}/F_{cr}$ | EI [MNm$^2$] |
|---|---|---|---|---|---|---|---|
| E_AF_01 | 9.4 | 19.5 | 13.6 | 34.8 | 2.6 | 2.1 | 1.261 |
| E_AF_02 | 7.1 | 16.6 | 12.6 | 45.0 | 3.6 | 2.3 | 1.028 |
| E_AF_03 | 12.0 | 15.5 | 21.6 | 31.0 | 1.4 | 1.3 | 1.014 |
| E_AF_04 | 13.2 | 15.8 | 21.8 | 29.0 | 1.3 | 1.2 | 1.105 |
| E_AF_05 | 11.9 | 12.5 | 21.1 | 22.5 | 1.2 | 1.1 | 1.030 |
| E_AF_06 | 16.0 | 18.2 | 23.7 | 30.0 | 1.1 | 1.1 | 1.123 |
| **Mean value** | **11.6** | **16.4** | **19.1** | **32.1** | **1.7** | **1.5** | **1.112** |

In the case of heat-strengthened TGIB-gf (Table 10), brittle fracture occurred without a warning signal despite the much higher load-bearing capacity. The maximum displacements (umax) are also quite similar. They average 41 mm for the silicone-bonded specimens, 40.6 mm for the acrylate-bonded specimens and at least 40.2 mm for the epoxy-bonded specimens. Again, it is expected that the flexural stiffness values are highest for the epoxy-bonded specimens, followed by acrylate and finally silicone.

**Table 10.** Results of measurements TGIB-gw with heat strengthened float glass—HS (E—epoxy, S—silicone, A—acrylate).

| Specimen | $F_{cr}$ = $F_{max}$ [kN] | $u_{cr}$ = $u_{max}$ [mm] | EI [MNm2] |
|---|---|---|---|
| S_HS_01 | 20.2 | 42 | 0.878 |
| S_HS_02 | 19.3 | 40 | 0.880 |
| **Mean value** | **19.8** | **41** | **0.879** |
| A_HS_01 | 25.1 | 39.1 | 1.171 |
| A_HS_02 | 25.2 | 42 | 1.095 |
| **Mean value** | **25.2** | **40.6** | **1.133** |
| E_HS_01 | 26.2 | 42 | 1.138 |
| E_HS_02 | 24.7 | 38.4 | 1.174 |
| **Mean value** | **25.5** | **40.2** | **1.156** |

Comparing the results of measurements between TGIB-fw and TGIB-gf with annealed float glass with epoxy adhesive (Table 11), we found that the average value of forces at the first crack ($F_{cr}$) are 1 kN higher for TGIB-gf than for TGIB-fw, and the maximum forces ($F_{max}$) are 4.3 kN higher. The average value of maximum displacements was also higher for TGIB-gf for 8.9 mm. However, we were surprised by the results for ductility, which was also higher for TGIB-gf, while bending stiffness was almost the same for both groups of specimens.

**Table 11.** Comparison of results of measurements for TGIB-fw and TGIB-gw with annealed float glass with epoxy adhesive.

| | $F_{cr}$ [kN] | $F_{max}$ [kN] | $u_{cr}$ [mm] | $u_{max}$ [mm] | Ductility d = $u_{max}/u_{cr}$ | q = $F_{max}/F_{cr}$ | EI [MNm$^2$] |
|---|---|---|---|---|---|---|---|
| TGIB-fw_E_AF | 10.6 | 12.1 | 19.0 | 23.2 | 1.3 | 1.2 | 1.096 |
| TGIB-gf_E_AF | 11.6 | 16.4 | 19.1 | 32.1 | 1.7 | 1.5 | 1.112 |

In Table 11 we see that the average values of the maximum forces are very comparable. However, the bending stiffness is expected to be higher for TGIB-gf.

Table 12 presents the results of the measurements for TGIB-fw with fully tempered glass (FT) and TGIB-gf with heat-strengthened float glass (HSG, E-epoxy, S-silicone, A-acrylate).

**Table 12.** Comparison of results of measurements for TGIB-fw with fully tempered glass (FT) and TGIB-gf with heat-strengthened float glass (HSG), E—epoxy, S—silicone, A—acrylate).

| Specimen | $F_{cr} = F_{max}$ [kN] | $u_{cr} = u_{max}$ [mm] | EI [MNm$^2$] |
|---|---|---|---|
| TGIB-fw_S_FT | 19.4 | 48.0 | 0.594 |
| TGIB-gf_S_HSG | 19.8 | 41 | 0.879 |
| TGIB-fw_A_FT | 26.7 | 42.7 | 0.898 |
| TGIB-gf_A_HSG | 25.2 | 40.6 | 1.133 |
| TGIB-fw_E_FT | 23.4 | 37.9 | 0.889 |
| TGIB-gf_E_HSG | 25.5 | 40.2 | 1.156 |

*4.2. Numerical Analysis*

Table 13 gives the results of the experimental analysis for the gamma coefficients of TGIB-fw and TGIB-gf. Considering Equations (16) and (17) from chapter 2 and the material properties of all the materials used the gamma coefficients can also be calculated.

**Table 13.** Comparison of results of measurements for TGIB.

| | Experimental Tests | | Gamma Method | |
| | Ɣ—TGIB-fw | Ɣ—TGIB-gf | Ɣ—TGIB-fw | Ɣ—TGIB-gf |
|---|---|---|---|---|
| Silicone | 0.577 | 0.720 | 0.794 | 0.681 |
| Acrylate | 0.932 | 0.907 | 0.989 | 0.981 |
| Epoxy | 0.922 | 0.898 | 0.999 | 0.999 |

From the comparison of the obtained results for the two test groups, TGIB-fw and TGIB-gf, the largest deviations between the results were for the most elastic adhesive, silicone, and the smallest was for the most rigid adhesive, epoxy. As expected, the highest values for both test groups were found for the samples bonded with epoxy adhesive. In the analytical analysis, the material properties for Sikasil SG500 silicone were adapted for TGIB-fw as specified by the manufacturer, and here the reason for the difference between the results can be found. The researcher who studied TGIB-gf also conducted tests with the silicone, using the material properties he himself determined. Regarding the results obtained with the acrylic, it should be noted that it is one of the stiffer acrylics on the market. The TGIB-fw samples had slightly higher values on average. The comparison of the obtained results showed that the type of glass does not have a significant influence on the bending stiffness of the whole timber-glass I-beam. The choice of adhesive has the greatest influence. The stiffer the adhesive chosen, the stiffer the entire structural element.

## 5. Discussion

Thus, if we compare the results between the epoxy bonded specimens TGIB-fw and TGIB-gf with annealed float glass, we can see that the results are quite comparable. AF glass shows a very brittle behaviour, that is, the relationship between stress and specific deformation is linear-elastic until the tensile strength of the material is reached, followed by an immediate loss of load-bearing capacity or failure. In a TGIB, this effect is seen when annealed float glass is used as a crack bridge that develops at the point of maximum tensile stress concentration and then grows in the appropriate direction as a function of subsequent stress distributions. Annealed float glass is more ductile because the appearance of cracks also predicts ultimate failure. TGIB-gf with this type of glass and epoxy adhesive shows an average 1 kN higher force value during the first crack formation compared to TGIB-fw. However, the behaviour of both groups of specimens during the test was similar. We can now conclude that the behaviour pattern of TGIBs with annealed float glass is linear until the first crack occurs. This fact must be taken into account when calculating the stiffness. It was expected that the bending stiffness would be higher for TGIB-gf, and it is. This

cross section basically offers higher stiffness, while TGIB-fw offers better behaviour against possible temperature variations. This is not possible with TGIB-gf. A comparison of the results of TGIB-fw shows that the specimens bonded with epoxy adhesive have the highest bending stiffness. Epoxy is also the stiffest adhesive compared to the silicone and acrylic adhesives used, so we were not surprised by the results. The specimens bonded with silicone adhesive, which is also the most elastic of the three adhesives, had the lowest stiffness. A correspondingly high stiffness was also achieved with acrylate, indicating that the acrylate used is one of the stiffer ones. Based on the above, it can be confirmed that the stiffness and load bearing capacity of TGIB depend on the type of glass web used.

An exact comparison between fully tempered glass and heat-strengthened float glass is not possible, mainly because of differences in the bending strength of the glass. The bending strength of toughened glass was 50 N/mm$^2$ higher. They have in common that the first crack was the end of the story. Both types of tempered glass are extremely fragile and fail without warning. We equated the first crack with failure. However, it can be seen that the fracture forces for the test specimens bonded with acrylate were highest for TGIB-fw with 26.7 kN and lowest for silicone, with an average of 19.4 kN. For TGIB-gf and HS glass, it was observed that the fracture forces for epoxy and acrylate are practically identical at 25 kN. The silicone specimens have on average a 5 kN lower fracture force, which was also reflected in a lower bending stiffness compared to the others. The stiffness of the TGIB-fw tested with fully tempered glass showed similar results to those with annealed float glass. Again, the highest stiffness values were obtained for the specimens bonded with acrylate and epoxy adhesives, while the lowest stiffness values were obtained for the specimens bonded with silicone adhesive.

Based on a study of the properties of the adhesives used and experimental work to determine the material properties, it can be confirmed that the stiffness and load-bearing capacity of I-beams made of timber and glass depend to a large extent on the choice and dimensions of the adhesive bond between the individual elements. It is worth considering that the performance of a TGIB system in static bonding largely depends on the reliability of the bond itself. This opens the way for further research, especially in terms of how to ensure the quality of this adhesive bond.

As expected, the subjects with the most elastic adhesive showed the highest displacements. In our case this was silicone, while the lowest displacements were naturally observed with the epoxy adhesive. In the case of fully tempered glass, no such large differences between the adhesives were observed. However, as already mentioned, the failure force was higher.

## 6. Conclusions

From the obtained results, it can be concluded that the cross section plays an important role in TGIBs made of annealed glass. For TGIB-fw, the first crack occured at a load 50% or more higher than for TGIB-gf. From both studies we can conclude that the bottom flange acts as a crack bridge which, together with the uncracked compression zone of the beam, ensured that the beam could continue to carry the load.

The timber–glass beam with fully tempered glass and heat-strengthened float glass failed immediately, but with a higher total load than the TG beams with annealed float glass. The results of the experimental analysis are very similar or almost identical for both cross sections (TGIB-gf and TGIB-fw). It can be concluded that the influence of the cross section is not as significant for these types of glass. Beams with cross-sections of heat-strengthened glass and fully tempered glass do not show any post-failure strength.

For TGIB-fw, we compared the ductility across the three test groups. The highest values were expected for silicone, which was also the most elastic adhesive. The values for epoxy were the lowest as expected due to the stiffness of the adhesive. For TGIB-gf, only the results for epoxy adhesive could evaluated. Here the ductility was slightly higher compared to the ductility of the epoxy adhesive samples of TGIB-fw. The reasons for this can be found in the material properties of the epoxy adhesives used. It can be seen that

the epoxy used in TGIB-gf was less rigid than the epoxy adhesive used in TGIB-fw. The TGIB-fw samples had slightly higher values for gamma coefficient, the highest values for both test groups were found for the specimens bonded with epoxy adhesive.

Further research could include numerical modelling of TGIB-fw. Experimental studies on TGIBs subjected to continuous long-term loading will be also interesting to investigate. This analysis would give us a more comprehensive insight into the behaviour of these beams, which are typically installed in buildings for a period of 50 years or more. Here, we could also investigate the influence of humidity and temperature, which would certainly yield interesting results. Then there is the area of vibrations, which is also an interesting research topic. In the future, life cycle analysis (LCA) could also be included, and testing could be carried out on test subjects using laminated glass.

**Author Contributions:** Conceptualization, M.D. and M.P.; methodology, M.D. and M.P.; software, M.D.; validation, M.D., M.P. and A.Š.; formal analysis, M.D.; investigation, M.D. and A.Š.; resources, M.D. and A.Š.; data curation, M.D.; writing—original draft preparation, M.D.; writing—review and editing, M.D., M.P. And A.Š.; visualization, M.D.; supervision, M.P. and A.Š.; project administration, M.P. and M.D.; funding acquisition, M.P. All authors have read and agreed to the published version of the manuscript.

**Funding:** This research received WoodWisdom-Net 2 research project LBTGC: URBAN WOOD, Wood based construction for multi-story, The potential of Application of Timber-Glass Composite Structures For Building Construction, no. 4302-7/2010/25, Acronym: LBTGC—Load bearing timber-glass composites.

**Institutional Review Board Statement:** Not applicable.

**Informed Consent Statement:** Informed consent was obtained from all subjects involved in the study.

**Data Availability Statement:** The data presented in this study are available on request from the corresponding author.

**Conflicts of Interest:** The authors declare no conflict of interest.

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
