# Peer review of "Influence of the Bonding Boundary Conditions of Timber-Glass I-Beams on Load-Bearing Capacity and Stiffness"

_applsci, doi:10.3390/app12041770_

Round 1

Reviewer 1 Report

The article contains very interested results suitable for publication. But from my opinion it is important to add the environmental condition during the preparing the samples that means the relative humidity and temperature of air, the relative humidity and temperature during the testing of samples and moisture of wood during the preparing the samples and during the testing of sample. into the article. The moisture of wood and relative humidity of air during testing and preparing samples have great impact on quality of results.

It would be good when the authors fill the units to describing mechanical properties and the ratio of misxing  adhesive and hardners to ahdhesives Sika SG 500 silicone  SikaFast 5215 NT .

Reviewer 2 Report

The manuscript investigated the effect of the cross-section of timber-glass I-beams on load-bearing capacity and stiffness. The manuscript is well organized and presented. The research methodology is scientifically sound. More references are suggested in the introduction section. In Section 4.2 - "Numerical analysis", more explanation is suggested, such as the numerical models applied, the software or algorithms, and the equations used. The authors should also analyze the numerical analysis results in Table 12 and compare the results with the experimental study. 

Reviewer 3 Report

The present paper presents a very interesting study regarding the influence of the cross-section of the flange, type of glass and adhesive over the load carrying capacity of the timber-glass I-beams. The results are consistent but the authors should clarify some aspects that could potentially count as significant setbacks:

  • the title does not accurately describe the research presented in the manuscript (actually, the adhesive and glass type exert a significant influence over the general load carrying capacity of the beams - at least as per my understanding of the paper);
  • the authors should include more information regarding the experimental setup and the testing procedures (e.g. for the TGIB-fw cross-section there are 18 test specimens - the exact number of specimens should be mentioned for each configuration; it is not clear why the TGIB-fw and TGIB-gf configurations were considered for the comparison).

Overall, the manuscript is well written and the research could have a direct influence on the market. Therefore, I highly encourage the authors to consider the above-mentioned aspects.
